# Factors Associated with Virological Failure in First-Line Antiretroviral Therapy in Patients Diagnosed with HIV-1 between 2010 and 2018 in Israel

**DOI:** 10.3390/v15122439

**Published:** 2023-12-15

**Authors:** Tali Wagner, Itzchak Levy, Daniel Elbirt, Eduardo Shahar, Karen Olshtain-Pops, Hila Elinav, Michal Chowers, Valery Istomin, Klaris Riesenberg, Dikla Geva, Neta S. Zuckerman, Marina Wax, Rachel Shirazi, Yael Gozlan, Natasha Matus, Shirley Girshengorn, Rotem Marom, Ella Mendelson, Orna Mor, Dan Turner

**Affiliations:** 1Faculty of Medicine, Tel-Aviv University, Tel Aviv 6997801, Israel; taliwagner@mail.tau.ac.il (T.W.); itsik.levy@sheba.health.gov.il (I.L.); chowersm@post.tau.ac.il (M.C.); ella.mendelson@sheba.health.gov.il (E.M.); dant@tlvmc.gov.il (D.T.); 2National HIV-1 and Viral Hepatitis Reference Laboratory, Chaim Sheba Medical Center, Ramat Gan 5262112, Israel; neta.zuckerman@sheba.health.gov.il (N.S.Z.); marina.wax@sheba.health.gov.il (M.W.); rachel.shirazi@sheba.health.gov.il (R.S.); yael.gozlan@sheba.health.gov.il (Y.G.); 3Infectious Disease Unit, Chaim Sheba Medical Center, Ramat Gan 5262112, Israel; 4Faculty of Medicine, Hadassah Braun School of Public Health & Community Medicine, The Hebrew University, Jerusalem 9112102, Israel; elbirtda@clalit.org.il (D.E.); hila.elinav@gmail.com (H.E.); 5Immunology, Kaplan Medical Center, Rehovot 76100, Israel; 6Immunology Unit, Rambam Health Care Campus, Haifa 3109601, Israel; ed_shahar@rambam.health.gov.il; 7Rappaport Faculty of Medicine, Institute of Technology, Technion, Haifa 3109601, Israel; 8Hadassah Medical Center, Jerusalem 9112102, Israel; kerenop@hadassah.org.il; 9Infectious Diseases, Meir Medical Center, Kfar Saba 4428164, Israel; 10Hillel Yaffe Medical Center, Hadera 38100, Israel; valeryi@hy.health.gov.il; 11Goldman Medical School, Faculty of Health Sciences, Ben-Gurion University of the Negev, Beersheba 8410501, Israel; klaris@bgu.ac.il; 12Infectious Disease Institute, Soroka Medical Center, Beersheba 84101, Israel; 13Integristat Ltd., Tel Aviv 69051, Israel; diklah@integristat.com; 14Ichilov Medical Center, Crusaid Kobler AIDS Center, Tel Aviv 6423906, Israel; hivlab@tlvmc.gov.il (N.M.); shirleygi@tlvmc.gov.il (S.G.); rotemma@tlvmc.gov.il (R.M.)

**Keywords:** HIV-1, virological failure (VF), first-line ART, adherence, CD4 trajectory curves, INSTI regimens, people living with HIV-1 (PLHIV)

## Abstract

Despite the progress in contemporary antiretroviral therapy (ART) and the continuous changes in treatment guidelines, virological failure (VF) is still an ongoing concern. The goal of this study was to assess factors related to VF after first-line ART. A longitudinal cohort retrospective study of individuals on first-line ART diagnosed with HIV-1 in 2010–2018 and followed-up for a median of two years was conducted. Demographics, baseline and longitudinal CD4 counts, treatment regimens, adherence and VF were recorded. The Cox proportional hazards regression and mixed models were used. A cohort of 1130 patients were included. Overall, 80% were males and 62% were Israeli-born individuals. Compared to individuals diagnosed in 2010–2014, when treatment was initiated according to CD4 levels, those diagnosed in 2015–2018 were older and had lower baseline CD4 counts. VF was recorded in 66 (5.8%) patients. Diagnosis with CD4 <200 cells/mmᶟ with AIDS-defining conditions (HR = 2.75, 95%CI:1.52–4.97, *p* < 0.001) and non-integrase strand transfer inhibitor regimens (non-INSTI, HR = 1.80, 95%CI:1.01–3.24, *p* = 0.047) increased VF risk. No impact of baseline resistance was observed. We concluded that the early detection of HIV-1 infection and usage of INSTI-based regimens are recommended to reduce VF.

## 1. Introduction 

In 2014, the Joint United Nations AIDS Program established global targets aimed at a global reduction in HIV-1 cases to the point that 90% of people living with HIV (PLHIV) should have been diagnosed in 2020, with 90% of those diagnosed being on antiretroviral therapy (ART) and 90% of those on ART being virally suppressed, with further increases to 95% for each by 2030 [1]. Indeed, globally, 89% of PLHIV were on ART and 93% were virally suppressed in 2022 [2]. An earlier study that assessed the cascade of HIV care in a cohort of newly diagnosed PLHIV in Israel [3] suggested that the linkage to care is still not optimal. It estimated that only 82.7% of all PLHIV have been diagnosed, of whom 86% have been treated, and that 93.1% of treated PLHIV had low or undetectable viral loads (VL < 200 copies/mL). 

Achieving and maintaining viral load suppression remain important indicators of successful response to anti-retroviral treatments (ART), even in the era of many improvements in ART efficacy and safety. For example, the approval of the first-generation integrase inhibitors INSTIs raltegravir and elvitegravir in 2007 and 2012, respectively, and the second-generation INSTIs dolutegravir and bictegravir in 2013 and 2018, respectively [4,5], has changed the recommendations for first-line therapy. Additionally, single-tablet regimens (STR) that simplify treatment and increase medication compliance were approved [6], and recommendations to initiate therapy immediately after diagnosis were introduced [7]. In Israel, the immediate initiation of therapy regardless of CD4 status has already been applied for many patients since 2015 [8], and immediate treatment initiation was approved for all newly diagnosed PLHIV in 2017 [9]. However, failures (VL > 200 copies/mL) in first-line treatment are still being recorded [3,10]. 

Differences in the baseline characteristics of the HIV-1 population have already been shown to affect treatment outcome. Diagnosis at younger age [5], with symptoms of acquired immunodeficiency syndrome (AIDS) or low CD4 counts (<200 cells/mm^3^) [11] concomitant with missing physician or clinical appointments and reduced adherence to ART, all contribute to virological failure (VF) [5,11,12,13]. Moreover, although most current ART have a high genetic barrier, infection with viruses with resistance mutations may still result in first-line treatment failure. Indeed, World Health Organization (WHO) guidelines (2021) still recommend expanding global HIV drug resistance prevention programs and that they should include resistance testing for monitoring and surveillance efforts [14]. In Israel, there have been several reports on continuous changes in the characteristics of newly diagnosed PLHIV in recent years [15,16]. Diagnosis at an older age is one issue, with low CD4 counts and a continuous increase in the proportion of the heterosexual risk group, all factors that may be related to VF, having been documented in more recent years. Moreover, the high prevalence (21.7%) of baseline drug resistance mutations characterized the newly diagnosed HIV-1 carriers [15], and it was partially related to infection in regions with generalized or concentrated HIV-1 epidemics (e.g., Sub-Saharan Africa and the former Soviet Union (FSU)) in countries with limited access to ART [17]. In this study, we followed-up, for a median of two years, a cohort of patients who had been diagnosed with HIV-1 after 2010 and who started first-line ART, aiming to assess treatment outcomes. 

## 2. Materials and Methods

### 2.1. Patients

We conducted a retrospective multi-site longitudinal cohort study using data retrieved from all HIV-1 regional clinical centers in Israel. All PLHIV above 15 years of age who had been diagnosed with HIV-1 between January 2010 and December 2018 and who started first-line therapy were followed-up until VF, when they either switched to second-line therapy or died. Patients <15 years of age, adults diagnosed in years other than 2010–2018, transgender people, non-Israeli citizens, and PLHIV who were not treated or lacked essential clinical data were excluded. The final cohort of 1130 PLHIV was selected based upon the relative proportion of PLHIV monitored at each of the eight medical centers. Demographic (sex, age, country of birth, mode of HIV-1 transmission), immunological (baseline CD4 counts cells/mm^3^), clinical (period of diagnosis, baseline HIV-1 viral load Log copies/mL, AIDS-defining diseases [18]), and virological data (HIV-1 subtype and drug pretreatment drug resistance mutations (DRM)) were recorded [19,20]. First-line ART regimens were recorded and longitudinal CD4 counts were collected. Information on adherence to therapy (categorized as “appropriate” or “poor”) was obtained from electronic medical files and cross-matched with the documentation of the medical staff in each of the medical centers. Poor adherence was defined by missing medications, discontinuation of ART or abusing treatments (defined as taking medication not according to a doctor’s instructions). VF was defined as ≥2 consecutive measurements of viral load > 200 copies/mL [21] >24 weeks following first-line ART initiation. 

### 2.2. Statistical Analysis and Ethical Approval

Statistical analysis was performed using R studio version 1.3.1093. The libraries used in this analysis included: “gt”, “gtsummary”, “dplyr”, “ggplot2”, “tidyverse”, ”lcmm”, “survival”, and “survminer”. We accounted for the change in treatment recommendations by comparing the factors associated with VF between 2010 and 2014, when treatment was initiated and when CD4 status was CD4 < 350 cells/mm^3^ or CD4 < 500 cells/mm^3^ [7], and between 2015 and 2018, when treatment was offered immediately regardless of CD4 status [8,9]. Fisher’s exact tests were used for comparing proportions of categorical variables, and Wilcoxon rank sum tests were used for comparing continuous variables between different categories. Cumulative probability for VF was calculated, and the differences between the Kaplan–Meier curves were assessed by a long-rank test at the two-, four- and six-year treatment follow-ups. Longitudinal CD4 trajectories were derived by a linear mixed effects model with random intercept and random slope. In addition, univariate and multivariate Cox regression models were applied to examine the hazard ratios (HR) and associations between the VF and other parameters. Additional variables were added to the Cox model to adjust for potential confounding effects. The proportionality of HR of the continuous variable (HIV-1 viral load) was assessed by Schoenfeld residuals. Statistical significance was defined as *p* < 0.05.

This retrospective study was approved by the Sheba institutional review board (5803-18-SMC) and subsequently by the institutional review boards of all participating HIV centers, as well as by the ethical committee of Tel Aviv University. Informed consent was waived. All data were anonymized, cross-matched and coded as dual-encoding on demand. 

## 3. Results

Epidemiology

Table 1 describes the characteristics of the study population and compares the two study periods (2010–2014 and 2015–2018). The median age of the cohort at diagnosis was 37 years (IQR 30–45), 80% were males, 62% were Israeli-born and the most prominent risk groups were MSM (53%) and heterosexual contacts (30%). As expected, the baseline characteristics of PLHIV that had been diagnosed and treated in 2015–2018 were different from those diagnosed in 2010–2014. The 2015–2018 group were diagnosed at an older age (median age 40 years (IQR 33–48) compared to 36 years (IQR 29–44), *p* < 0.001), with lower baseline CD4 counts (322 (IQR 149–494) cells/mm^3^ versus 367 (IQR 183–544) cells/mm^3^, *p* = 0.045) and with a higher median baseline log VL (5.0 copies/mL (IQR 4.3–5.5) vs. 4.8 copies/mL (IQR 4.1–5.3), *p* = 0.003). The relative proportion of risk groups for HIV infection was also different. More infections among PLHIV occurred through heterosexual transmission (93/259, 35.9% vs. 246/871, 28.2%, *p* = 0.022) and more of the patients had non-B subtypes (158/259, 61% vs. 433/871, 49.7%, *p* = 0.003). Pretreatment resistance testing was performed in 85% of the entire cohort and >20% of them had baseline DRM.

Treatment regimens and the time that had elapsed between diagnosis and initiation of ART changed during the studied years. There was a significant annual increase in INSTI usage and a significant annual decrease in non-INSTI (*p* < 0.001, Figure 1a). Moreover, the proportion of INSTI-STR prescribed for 18.1% (205/1130) of the entire cohort also increased significantly during the more recent years (*p* < 0.001, Figure 1b). Treatment initiation was also significantly different between the two time periods: the time between diagnosis and first-line treatment was a median of 0.90 months (IQR 0.30–2.35) in 2015–2018 compared to 3.4 months in 2010–2014 (IQR 1.05–10.70), *p* < 0.001. The comparison between the median time elapsed between diagnosis and first-line treatment in the studied years, the common drug regimens and the proportion of STR usage is presented in Figure A1, and in Table A1 and Table A2 in the Appendix A section.

Table 2 compares the characteristics of those who failed treatment to those who did not. Overall, 5.8% (66/1130) had VF during a median follow-up time of 2.32 (IQR 0.94–4.12) years after first-line therapy. These individuals had significantly lower baseline CD4 counts (median 223 (IQR 101–404) cells/mm3 vs. 364 (IQR 183–543) cells/mm^3^, *p* = 0.001) were more likely to be diagnosed with Stage 3C (28.8%, 19/66 vs. 11.6%, 123/1064, *p* < 0.001), and were more likely to be infected with subtype C (25.8%, 17/66 vs. 11.2%, 119/1064, *p* = 0.001). During follow-up, poor treatment adherence was recorded in 16% (175/1130) of the study cohort, characteristic of those with VF (63.6%, 42/66). There was a higher proportion of non-INSTI regimens (56.1%, 37/66 vs. 42.7%, 454/1064, *p* = 0.062) and a lower number of STR regimens (9.1%, 6/66 vs. 18.7%, 199/1064, *p* = 0.072) in those with VF.

The cumulative probability of VF was higher in the non-INSTI regimens compared to the first and second INSTI regimens (*p* = 0.0024) and increased during the later years of follow-up (Figure 2). The VF rates after two, four and six years of treatment follow-up in patients on second-generation INSTI were 2% (CI 95 0–4%), 4% (CI 95 1–6%) and 10% (CI 95 2–17%), respectively, compared to the VF rates in non-INSTI regimens of 7% (CI 95 5–10%), 12% (CI 95 8–16%) and 17% (CI 95 10–23%), respectively (*p* = 0.0024). Moreover, assessment of longitudinal CD4 counts revealed different CD4 trajectories in patients with VF (*p* < 0.001). Patients who sustained VF had a more moderate pattern of increase in CD4 trajectory compared to patients who did not fail therapy (Figure A2).

A multivariate analysis (Table 3) revealed that HIV-1 stage 3C at diagnosis (CD4 < 200 cells/mmᶟ and AIDS-defining disease) and non-INSTI regimens were risk factors for VF. Risk group (hetero), birthplace (Israel), HIV-1 subtype (C), usage of STR or the NRTI M184V/I (not shown), which were all found to be associated with VF in the univariate analysis, did not remain related factors in the multivariate analysis after controlling for confounders. Contrarily, analysis of the treatment regimen and baseline DRM of 14 patients who failed first-line ART (Table A3) identified three cases of patients who received drugs to which they had high-level resistance mutations (with a total score of ≥60 [22]).

## 4. Discussion

This study followed up patients diagnosed with HIV-1 during 2010–2018 who started first-line therapy. Their characteristics and the main factors related to treatment success or failure were assessed. Recommendations regarding the initiation of ART have changed during the study years. While treatment in 2010–2014 was initiated based upon CD4 counts, not only were second-generation INSTI already widely used, but ART was also offered to newly diagnosed individuals regardless of the CD4 levels since 2015 [7,8]. With such a rapid initiation of novel INSTI-including regimens (less than a median of one month after initial diagnosis), a reduction in the prevalence of treatment failures could be expected. Indeed, the cumulative probability of VF in patients on second-generation INSTI was much lower than that of those treated with non-INSTI (*p* = 0.0024); however, VF on new INSTI could still be observed. Patients in the VF group were further characterized by lower baseline CD4 levels compared to those in the non-VF group, although both groups had CD4 counts related to late diagnosis (median of 223 versus 364 cells/mm^3^, respectively, *p* < 0.001) [23,24]. Moreover, PLHIV who were diagnosed with the most advanced stage of HIV-1 (CD4 < 200 cells/mm3 and AIDS-defining conditions) were 2.75 times more at risk for VF compared to other patients. Late diagnosis has been connected to difficulties in access to and uptake of HIV testing in many countries [25], and this characterized the majority of PLHIV (53%) in Europe in 2019 [26]. 

Univariate analysis of longitudinal CD4 counts revealed a more rapid increase in CD4 trajectory in patients with no VF (*p* < 0.001). This is in line with the suggestion that an increase in CD4 cell count is extremely rapid in individuals who can avert viral loads >1000° copies/mL while on ART [27]. 

Adherence is well-documented as a major determinant of HIV treatment outcomes [28]. PLHIV experiencing VF reported worse adherence over time compared with those not failing therapy, indicating that non-adherence to therapy remains a concern despite many innovations and immediate ART initiation. A recent report from the United States demonstrated that a 10% increase in adherence increased the odds of being virally suppressed by 37% [29]. 

Although maximum adherence to ART is essential, a recent prospective multicenter cohort study suggested that each class of antiretroviral drugs has specific levels and ranges of adherence–HIV–RNA relationship, which allows one regimen to be more flexible than others [30]. For example, dolutegravir was reportedly more “forgiving” of missed doses compared to older ART, and the risk of HIV replication was shown to have been reduced with that second-generation INSTI [30]. Our results corroborate with the superiority and safety of INSTI regimens compared to non–INSTI ones. The inferiority of non-INSTI regimens may be partially explained by their lower efficacy [31] or, alternatively, by the reduced tolerability to some of those regimens [32].

The use of STR decreased the risk of VF in the univariate analysis, and did so in the multivariate model, although the results did not reach a level of significance (*p* = 0.05), as demonstrated in other studies as well [6]. The most common STRs were the combination of dolutegravir, abacavir and lamivudine in a single pill (DTG, ABC, 3TC) and the combination of rilpivirine, tenofovir and emtricitabine (RPV, TDF, FTC), each constituting 24.4% of all STRs. Importantly, the DTG, ABC, and 3TC types of STR must be taken around the same time every day [33] and the RPV, TDF, and FTC types of STR must be taken with food [34]. All of these requirements suggest that patients should be aware of adverse effects and perform necessary lifestyle adjustments to improve treatment success, even in simplified treatments such as STR.

There was a difference in the proportion of baseline NRTI DRM in the PLHIV with VF (*p* = 0.045), particularly in those with M184IV (*p* = 0.015), in the univariate analysis. However, this effect was lost in the multivariate analysis. Also, no significant association was found between having resistant tests at baseline and between VF adjusted to other characteristics. Although no clinical relation was found between the performance of a baseline resistance test and the VF, the results of such testing provide the clinician with useful information. Resistance testing may define subtypes of the HIV-1 virus, and some subtypes may be more virulent than others [35,36] and thus impact treatment decisions. Moreover, integrase resistance testing should be routinely performed in the era of INSTI-based regimens. Baseline integrase resistance test results were available for only 17% (163/960) of our cohort, and no major baseline resistance to INSTI was recorded. 

The finding of treatment adjustment in cases with resistance is also worth noting. Three of fourteen patients who failed first-line ART did not receive the most appropriate treatment. The first patient who had baseline resistance K103NS, which decreases susceptibility to efavirenz (EFV), was treated with EFV, TDF and FTC. The second patient who had M184IV (which may decrease the susceptibility of 3TC) was treated with raltegravir (RAL), ABC and 3TC. Similarly, another patient with M184IV (affecting FTC) received RAL, TDF and FTC [22]. Stanford HIVdb, which has the potential drug effects of other DRMs, may assist in clinical decision-making when a resistant profile is known, thereby increasing potential treatment success [37].

Our study had several major limitations. First, we could not measure treatment adherence directly since the drug blood level, which is the most appropriate objective measurement, is not routinely assessed in most countries, and no drug-level tests whatsoever are performed in Israel. Pharmacy dispensing can also be used to measure adherence, but it may also be inappropriate since some patients may not take ART even when prescribed. Indeed, measuring adherence is regarded problematic [28]. To partially overcome this obstacle, we defined treatment adherence based upon medical electronic files and cross-matched the data with the documentation of the medical staff in each of the medical centers. Second, not all patients diagnosed in 2010–2018 were included in this study. For example, we excluded undocumented immigrants who have limited access to healthcare due to lack of medical insurance. Prescription of ART, start treatment guidelines, schedule of clinical follow-up visits and availability of monitoring tests are different in undocumented immigrants, and therefore their follow-up outcome cannot be compared to that of HIV patients with Israeli citizenship [38].

In conclusion, despite the increase in the use of potent regimens, VF continues to be a matter of concern. Public health efforts should focus upon PLHIV with poorer treatment adherence, those on non-INSTI treatment regimens, and those with a low baseline starting point of CD4 counts (<200) and a moderate longitudinal increase in CD4 trajectory. 

## Figures and Tables

**Figure 1 viruses-15-02439-f001:**
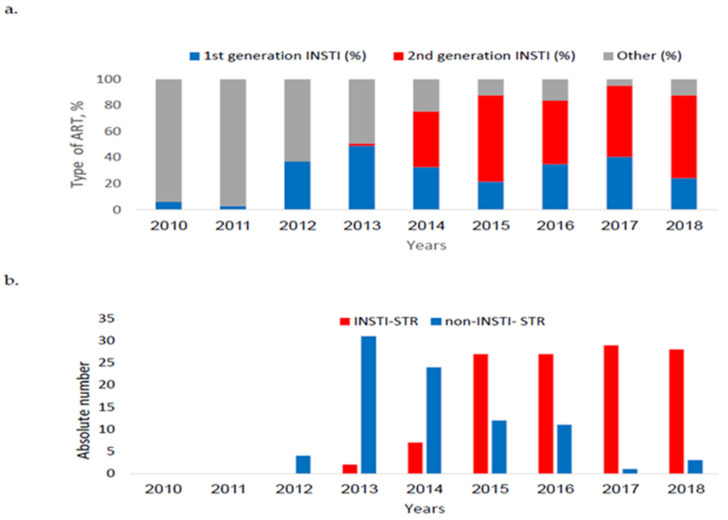
First-line regimens in 2010–2018. (**a**) Proportion of 1st generation INSTI, second-generation INSTI and non-INSTI regimens (other); (**b**) Number of INSTI-STR and non-INSTI-STR. INSTI-integrase strand transfer inhibitors; STR-single tablet regimen.

**Figure 2 viruses-15-02439-f002:**
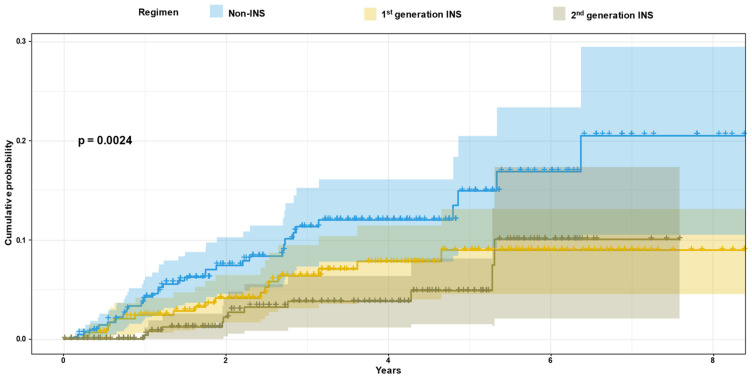
Cumulative probability of VF by treatment regimens. INSTI-integrase strand transfer inhibitors. The blue solid line shows the cumulative probability of VF, and blue shadowing shows the 95% CI (confidence interval) of the non-INSTI regimens. The yellow solid line shows the cumulative probability of VF, and yellow shadowing shows the 95% CI of the first-generation INSTI. The grey solid line shows the cumulative probability of VF, and grey shadowing shows the 95% CI of the second-generation INSTI.

**Table 1 viruses-15-02439-t001:** Characteristics of the study cohort (N = 1130) compared by years of diagnosis.

Characteristics	Overall, N = 1130	2010–2014 N = 871 (77.1%)	2015–2018 N = 259 (22.9%)	pv
Age, median (IQR), years	37 (30–45)	36 (29–44)	40 (33–48)	<0.001
CD4 baseline count, median (IQR), cells/mm^3^	357 (178–535)	367 (183–544)	322 (149–493)	0.045
HIV-1-RNA baseline, median (IQR), log copies/mL	4.8 (4.2–5.4)	4.8 (4.1–5.3)	5.0 (4.3–5.5)	0.003
Sex				
Female, n (%)	222 (19.6)	170 (19.5)	52 (20.1)	0.913
Male, n (%)	908 (80.4)	701 (80.5)	207 (79.9)	
Birthplace				0.058
IL, n (%)	695 (61.5)	526 (60.4)	169 (65.3)	
SSA, n (%)	133 (11.8)	111 (12.7)	22 (8.5)	
EEU/CA, n (%)	211 (18.7)	155 (17.8)	56 (21.6)	
WCEU/NA, n (%)	41 (3.6)	37 (4.2)	4 (1.5)	
LA, n (%)	17 (1.5)	14 (1.6)	3 (1.2)	
ME/NA	25 (2.2)	20 (2.3)	5 (1.9)	
AP	8 (0.7)	8 (0.9)	0	
HIV-1 Stage 3c, n (%)	142 (12.5)	99 (11.4)	43 (16.6)	0.034
Transmission				0.071
Hetero, n (%)	339 (30)	246 (28.2)	93 (35.9)	
MSM, n (%)	596 (52.7)	464 (53.3)	132 (51)	
Bisexual, n (%)	81 (7.2)	64 (7.3)	17 (6.6)	
IDU, n (%)	51 (4.5)	47 (5.4)	4 (1.5)	
IDU-Hetero, n (%)	43 (3.8)	33 (3.8)	10 (3.9)	
IDU-MSM, n (%)	12 (1.1)	10 (1.1)	2 (0.8)	
Other/Unknown	8 (0.7)	7 (0.8)	1 (0.4)	
Subtype				0.001
A1, n (%)	31 (2.7)	20 (2.3)	11 (4.2)	
A6, n (%)	162 (14.3)	124 (14.2)	38 (14.7)	
CRF01_AE, n (%)	18 (1.6)	15 (1.7)	3 (1.2)	
CRF02_AG, n (%)	28 (2.5)	20 (2.3)	8 (3.1)	
B, n (%)	539 (47.7)	438 (50.3)	101 (39)	
C, n (%)	136 (12)	109 (12.5)	27 (10.4)	
Other, n (%)	74 (6.5)	46 (5.3)	28 (10.8)	
Unknown, n (%)	142 (12.6)	99 (11.4)	43 (16.6)	
Basic Resistance Test, n (%)	960 (85)	746 (85.6)	214 (82.6)	0.237
Baseline DRM, (n = 960), (85%)	200 (20.8%)	148 (19.8%)	52 (24.3%)	0.181
NNRTI, n (%)	116 (12.1)	87 (11.7)	29 (13.6)	0.453
E138AKQ, n (%)	58 (6)	48 (6.4)	10 (4.7)	0.434
K103NS, n (%)	50 (5.2)	36 (4.8)	14 (6.5)	0.406
NRTI, n (%)	46 (4.8)	37 (5)	9 (4.2)	0.630
M184IV, n (%)	5 (0.5)	5 (0.7)	0	0.509
PI, n (%)	56 (5.8)	39 (5.2)	17 (7.9)	0.181
L90M, n (%)	27 (2.8)	19 (2.5)	8 (3.7)	0.483

n—number; pv—*p* value; IQR—interquartile range; IL—Israel; SSA—Sub-Saharan Africa; EEU/CA—Eastern Europe and Central Asia; WCEU/NA—Western and Central Europe and North America; LA—Latin America; ME/NA—Middle East and North Africa; AP—Asia and Pacific; Hetero—heterosexual; MSM—men who have sex with men; IDU—injecting drug users; HIV-1 Stage 3c—CD4 counts <200 cells/mm^3^ and AIDS defining diseases; DRM—drug resistance mutations; NNRTI—non-nucleoside reverse transcriptase inhibitors; NRTI—nucleoside reverse transcriptase inhibitors; PI—protease inhibitors; INSTI—integrase strand transfer inhibitors.

**Table 2 viruses-15-02439-t002:** Characteristics of the study cohort categorized as VF and non-VF (n = 1130).

Characteristics	Overall, N = 1130	No VF N = 1064 (94.2%)	VF, N = 66 (5.8%)	pv
Number of follow-up years, median (IQR)	2.32 (0.94–4.12)	2.38 (0.96–4.21)	1.56 (0.77–2.69)	0.006
Age, median (IQR), years	37 (30–45)	37 (30–45)	39 (31–45)	0.449
CD4 baseline count, median (IQR), cells/mm^3^	357 (178–535)	364 (183–543)	223 (101–404)	0.001
HIV-1-RNA baseline, median (IQR), log copies/mL	4.8 (4.2–5.4)	4.8 (4.2–5.4)	4.9 (4.1–5.5)	0.617
Sex				
Female, n (%)	222 (19.6)	205 (19.3)	17 (25.8)	0.259
Male, n (%)	908 (80.4)	859 (80.7)	49 (74.2)	
Birthplace				
IL, n (%)	695 (61.5)	665 (62.5)	30 (45.5)	0.009
HIV-1 Stage 3c, n (%)	142 (12.5)	123 (11.6)	19 (28.8)	<0.001
Transmission				
Hetero, n (%)	339 (30)	312 (29.3)	27 (40.9)	0.064
Subtype				
C, n (%)	136 (12)	119 (11.2)	17 (25.8)	0.001
Years of diagnosis				
2010–2014, n (%)	871 (77.1)	814 (76.5)	57 (86.4)	0.089
2015–2018, n (%)	259 (22.9)	250 (23.5)	9 (13.6)	
Basic resistance test, n (%)	960 (85)	908 (85.3)	52 (78.8)	0.205
DRM baseline, (N = 960), n (%)	200 (20.8)	186 (20.5)	14 (26.9)	0.349
NNRTI, n (%)	116 (12.1)	108 (11.9)	8 (15.4)	0.346
E138AKQ, n (%)	58 (6)	56 (6.2)	2 (3.8)	0.701
K103NS, n (%)	50 (5.2)	46 (5.1)	4 (7.7)	0.611
NRTI, n (%)	46 (4.8)	40 (4.4)	6 (11.5)	0.045
M184IV, n (%)	5 (0.5)	3 (0.3)	2 (3.8)	0.015
PI, n (%)	56 (5.8)	55 (6)	1 (1.9)	0.351
L90M, n (%)	27 (2.8)	26 (2.9)	1 (1.9)	1.000
Treatment regimens				
First-generation INSTI, n (%)	331 (29.3)	313 (29.4)	18 (27.3)	0.062
Second-generation INSTI, n (%)	308 (27.3)	297 (27.9)	11 (16.7)	
Non-INSTI, n (%)	491 (43.4)	454 (42.7)	37 (56.1)	
STR, n (%)	205 (18.1)	199 (18.7)	6 (9.1)	0.072
Adherence (poor), n (%)	175 (15.5)	133 (12.5)	42 (63.6)	<0.001

n—number; VF—viral failure; pv—*p* value; INSTI—integrase strand transfer inhibitors; IQR—interquartile range; HIV-1 Stage 3c at diagnosis—CD4 counts <200 cells/mm^3^ and AIDS-defining disease; DRM—drug resistance mutations; NNRTI—non-nucleoside reverse transcriptase inhibitors; NRTI—nucleoside reverse transcriptase inhibitors; PI—protease inhibitors; INSTI—integrase strand transfer inhibitors; SRT—single tablet regimen.

**Table 3 viruses-15-02439-t003:** Association of demographics and clinical data with virological failure of the study cohort (N = 1130).

Characteristics	Unadjusted HR (95%CI), *p*	Adjusted HR (95%CI), *p*
Age (years)	1.00 (0.98–1.03), 0.793	
CD4 baseline count, cells/mm^3^	1.00 (1.00–1.00), <0.001	
HIV-1-RNA baseline (log copies/mL)	1.12 (0.86–1.45), 0.400	0.98 (0.76–1.28), 0.901
Sex		
Female	Ref	
Male	0.70 (0.40–1.21), 0.203	
Birthplace (Israel versus other)	0.53 (0.33–0.87), 0.011	0.75 (0.42–1.34), 0.327
HIV-1 Stage (3c versus other)	3.39 (1.99–5.77), <0.001	2.75 (1.52–4.97), <0.001
Transmission (hetero versus other)	1.65 (1.01–2.70), 0.045	0.79 (0.41–1.51), 0.476
Subtype (C versus other)	2.12 (1.22–3.69), 0.008	1.70 (0.89–3.25), 0.108
Baseline resistance test	0.58 (0.32–1.04), 0.066	0.59 (0.31–1.12), 0.105
Years of diagnosis, (2015–2018)	0.66 (0.32–1.33), 0.244	1.09 (0.48–2.49), 0.832
Treatment regimens		
First generation INSTI	Ref	
Second generation INSTI	0.65 (0.31–1.37), 0.257	0.65 (0.29–1.42),0.277
Non-INSTI regimens	1.86 (1.06–3.27), 0.031	1.80 (1.01–3.24), 0.047
Single-tablet regimen	0.43 (0.18–0.99), 0.050	0.50 (0.20–1.19), 0.119

All of the factors were included in the multivariate analysis with the exceptions of age, sex and baseline CD4 counts, which were removed either due to their non-significance in unadjusted HR (age and sex), no difference in the spread of the IQR range (age), a small sample size of females (sex), or to prevent the correlation with the HIV-1 Stage 3c (baseline CD4 counts <200 cells/mm^3^ and AIDS-defining disease). Time-variable covariates such as adherence were also removed from the cox analysis. Proportionality of hazard ratios of the continuous variable (HIV-1 viral load) were assessed by Schoenfeld residuals (Figure A3). HR—hazard ratio; CI—confidence interval; *p*—*p* value; INSTI—integrase strand transfer inhibitors.

## Data Availability

Raw data can be provided upon reasonable request to the authors.

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
