# Peer review of "Factors Associated with Virological Failure in First-Line Antiretroviral Therapy in Patients Diagnosed with HIV-1 between 2010 and 2018 in Israel"

_viruses, 2023, doi:10.3390/v15122439_

Round 1

Reviewer 1 Report

Comments and Suggestions for Authors

The work by Wagner and colleagues  analyzes the epidemiology and risk factors of treatment failures of first-line regimen prescribed in Israel between 2010 and 2018, focusing also on the different "eras" of ARV regimens. The work is well written and the statistical analysis is on point. 

My first comment is that the definition of virological failure, although well described in the Methods, appears to be not too rigorous. A cut-off of 200 copies in 2023 in a Western country may appear a little outdated given the availability of highly efficacy drugs.  

I would suggest to revise and substitute "patients" with "PLWHIV" (i.e. line 105)

Comments on the Quality of English Language

Good English. May benefit from minor improvements.

Author Response

  1. My first comment is that the definition of virological failure, although well described in the Methods, appears to be not too rigorous. A cut-off of 200 copies in 2023 in a Western country may appear a little outdated given the availability of highly efficacy drugs.  

Answer: Thank you very much for your comment. Indeed the definition of viral failure is being continuously assessed and updated. In our study we focused on PLHIV who were diagnosed in 2010-2018, and many of them initiated ART with non-INSTI regimens (43.4%) and early generation INSTI (29.3%), therefore we relied on more rigorous definitions as suggested also by the American guidelines for the use of ART in Adult and Adolescents living with HIV, with a cut-off 200 copies (1). Importantly, after identifying virologic failure, or in cases in which there is concern for incomplete virologic response, resistance testing is suggested and required to determine DRM. When viral load is less than 200 copies/mL some labs do not preform resistance testing and in many cases, resistance tests failed technically resulting in lack of relevant information for our study. Therefore, a cut-off of 200 copies was chosen.

  1. https://clinicalinfo.hiv.gov/en/guidelines/hiv-clinical-guidelines-adult-and-adolescent-arv/virologic-failure?view=full

  1. I would suggest to revise and substitute "patients" with "PLWHIV" (i.e. line 105)

Answer: Thank you, was done. We revised "patients" with "PLHIV".

Revised version: The final cohort of 1130 PLHIV was randomly selected based on the relative proportion of PLHIV monitored at each of the eight medical centers.

  1. Good English. May benefit from minor improvements.

Answer: Thank you. Done

Reviewer 2 Report

Comments and Suggestions for Authors

This manuscript by Wagner et al. is a retrospective study that included HIV-infected individuals who were on first-line ART and diagnosed in 2010-2018. These patients were followed up at two different time intervals (2010-2014 and 2015-2018) i.e., a median of two years. Different statistical analyses were conducted to determine the relationship between viral failure (VF), time of first-line ART, and treatment regimen. The authors concluded that detection of HIV-1 infection together with adherence to single-tablet regimen and integrase strand transfer inhibitors (INSTIs)-based regimens and monitoring are recommended to reduce VF.

As such, This study or the conclusions drawn are not new. There are numerous reports, which more or less, have deduced similar conclusions. Having said that, this study is important in the context of the resourceful country of Israel. It is surprising to note that Israel could not achieve the 90/90/90 goal despite all available resources. The results presented in this study will most certainly help in assessing the HIV-1 infection status of Israel.

I have only two main concerns. This is about the methods used in the study. The authors state that they used RStudio for statistical analyses. To the best of my knowledge, Rstudio is a GUI that can interface with different programming languages. It is recommended that the authors specify what libraries (of R or Python) were used for the analyses. 

The other concern is that the authors sis not discuss the confounding factors in their study. 

Once, these concerns are addressed, I believe that the MS will be in good shape.

Author Response

  1. The authors state that they used RStudio for statistical analyses. It is recommended that the authors specify what libraries (of R or Python) were used for the analyses. 

Answer: Thank you, the specified libraries were added in yellow.

 Statistical analysis was performed using R studio version 1.3.1093. The libraries which were used in this analysis included: library("gt"), library("gtsummary"),library("dplyr"),library("lcmm"),library("ggplot2"),library("tidyverse"),library("survival"),library("survminer").

We revised the text in methods section accordingly

  1. The other concern is that the authors sis not discuss the confounding factors in their study. 

Answer: Thank you. We added in the methods (line 134-135): Potential confounders were controlled by the adjusted Cox regression analysis.

Moreover, we added in the results (line 221-222): Multivariate analysis (Table 3) revealed that, unlike in the univariate analysis, later period of diagnosis, 2015-2018, became a significant risk factor to VF (after controlling for confounders).

Line (227-230) :Risk group (Hetero), birthplace (Israel), HIV-1 subtype (C) and baseline resistance test, or the NRTI, including M184V/I (not shown), which were all found to be associated with VF in univariate analysis, did not remain related factors in multivariate analysis (after controlling for confounders
